# The Depressiveness, Quality of Life and NEO-FFI Scale in Patients with Selected Genodermatoses

**DOI:** 10.3390/jcm13061624

**Published:** 2024-03-12

**Authors:** Bartlomiej Wawrzycki, Magdalena Fryze, Radosław Mlak, Alicja Pelc, Katarzyna Wertheim-Tysarowska, Anette Bygum, Aleksandra Wiktoria Kulbaka, Dariusz Matosiuk, Aldona Pietrzak

**Affiliations:** 1Department of Dermatology, Venereology and Paediatric Dermatology, Medical University of Lublin, 20-080 Lublin, Poland; bartekwawrzycki@gmail.com; 2Department of Psychology, Psychosocial Aspects of Medicine, Medical University of Lublin, 20-059 Lublin, Poland; magdalena.fryze@umlub.pl; 3Department of Laboratory Diagnostics, Medical University of Lublin, 20-059 Lublin, Poland; radoslawmlak@umlub.pl; 4Psychology and Psychotherapy Center Euro-Medica, Garbarska 20, 20-340 Lublin, Poland; alapelc1996@gmail.com; 5Non-Public Kindergarten Poziomkowa Academy, 23-100 Bychawa, Poland; 6Department of Medical Genetics, Institute of Mother and Child, 01-211 Warsaw, Poland; katarzyna.wertheim@imid.med.pl; 7Clinical Institute, University of Southern Denmark, 5230 Odense, Denmark; anette.bygum@rsyd.dk; 8Department of Clinical Genetics, Odense University Hospital, 5000 Odense, Denmark; 9Independent Researcher, London NW8 8NJ, UK; awkulbaka1@gmail.com; 10Department of Synthesis and Chemical Technology of Pharmaceutical Substances, Faculty of Pharmacy, Medical University of Lublin, 20-059 Lublin, Poland; dariusz.matosiuk@umlub.pl

**Keywords:** Beck’s Depression Inventory, NEO Five-Factor Inventory, Dermatology Life Quality Index, genodermatosis, ichthyosis, palmoplantar keratoderma

## Abstract

**Background:** Dermatological conditions extend beyond physical symptoms, profoundly impacting the psychological well-being of patients. This study explores the intricate relationship between depressive symptoms, quality of life (QoL), and personality traits in individuals diagnosed with specific genodermatoses. **Methods:** The study cohort comprised 30 patients with genodermatoses treated at the dermatology clinic, and a healthy control group. Standardized survey questionnaires: The Dermatology Life Quality Index (DLQI), Beck’s Depression Inventory (BDI), and NEO Five-Factor Inventory (NEO-FFI) were employed for assessments. **Results:** The findings indicate a significantly elevated risk of severely or very severely reduced QoL in the study group compared to matched controls (OR = 22.2, 95% CI: 2.7–184.8). Specifically, individuals with ichthyosis exhibited a staggering 131-fold higher risk of diminished QoL compared to the control group. Furthermore, the prevalence of depression was higher in the study group than in the control group (36.7% vs. 10%; *p* = 0.0086). A detailed analysis revealed that patients with low or average agreeableness exhibited a notably higher incidence of depression compared to those with high agreeableness (100% or 75% vs. 28.6%; *p* = 0.0400). Similarly, individuals with high levels of neuroticism had a significantly higher incidence of depression compared to those with average or low levels of neuroticism (rates: 66.7% vs. 9.1% or 0%, respectively; *p* = 0.0067). **Conclusions:** The study underscores a substantial correlation between genodermatoses and the mental health of affected individuals, underscoring the imperative consideration of psychological factors in the management of hereditary skin disorders. Our study’s primary limitation is the small sample size, stemming from difficulties in recruiting participants due to the rare nature of the studied conditions.

## 1. Introduction

The impact of dermatological conditions reaches beyond the physical symptoms, significantly affecting patients’ psychological health. The presence of characteristic skin lesions, particularly on exposed parts of the body, can lead to isolation, irritability, sadness, and low self-esteem in patients. The affected individuals may also experience physical symptoms such as pruritus, pain, desquamation, skin redness, and, notably in certain types of ichthyoses, complications like anhidrosis, alopecia and ectropion. Furthermore, patients with certain dermatological conditions are sometimes excluded from the community. These aspects can affect the ability of patients to work, learn and participate in social life [1,2,3,4]. 

The psychological impact of dermatological conditions has been measured using various tools. Quality of life (QoL) describes the overall well-being of individuals or entire societies. It includes multiple domains such as physical and mental health, employment, education, identity, or wealth [5]. The Dermatology Life Quality Index (DLQI) is a questionnaire that addresses skin issues in various aspects of life [6]. Another tool is the NEO Five-Factor Inventory (NEO-FFI), which is a scale for measuring five key personality aspects: neuroticism, extraversion, openness to experience, agreeableness, and conscientiousness. The presence and severity of depressive symptoms may be assessed using Beck’s Depression Inventory (BDI) and modified version BDI-II [7].

The existing research has unequivocally demonstrated that there is a strong association between dermatological conditions and mental health problems, with mental disorders affecting between 30% and 40% of dermatological patients [1,2,3,4]. Depression often accompanies inflammatory skin disorders like atopic dermatitis and psoriasis [8]. Numerous studies have examined the influence of ichthyosis on the quality of life, some specifically addressing the QoL of children with congenital ichthyosis. The findings of these studies indicate that congenital ichthyosis can adversely affect various aspects of life, leading to a diminished QoL [5,9,10]. 

The aim of this study was to investigate the interplay between depressive symptoms, QoL, and personality traits assessed by the NEO-FFI Scale in patients with selected genodermatoses. Through a comprehensive analysis, we sought to determine the impacts of these dermatological conditions on mental health, with a particular focus on understanding the association between specific genodermatoses, overall QoL, and the occurrence of depression and its levels. Additionally, we aimed to explore the role of personality traits, as measured by the NEO-FFI Scale, in influencing the psychological well-being of individuals affected by genodermatoses. 

## 2. Materials and Methods

### 2.1. Study Population

The study population included patients with genodermatoses, treated at the Dermatology Outpatient Langiewicza 6a in Lublin and Dermatology Clinic of the Clinical Public Hospital No. 1 in Lublin. They were enrolled in the study between April 2019 and September 2023. Most patients had their diagnoses verified genetically based on genetic testing performed at the Institute of Mother and Child, Warsaw or MedGen Medical Center, Warsaw. All the patients provided informed consent to join the study. The Bioethics Committee at the Medical University of Lublin approved the study (KE-0254/91/2021, KE 0254/87/03/2023). 

The control group was recruited from healthy hospital employees and patient family members without skin disorders who belonged to the same cultural circle. The study and control groups were European Caucasians.

### 2.2. Patients Assessment

The author’s questionnaire included inquiries regarding sociodemographic details and specific questions about the symptoms associated with ichthyosis. The first part contained standard variables such as gender, age, education, and residency. The second focused on symptoms related to the disease, such as pain, scaling, itching, peeling, redness, infiltration, and the affected area. 

The study was based on the standardized survey questionnaires DLQI, BDI and NEO-FFI. The QoL of patients with ichthyosis was assessed using the DLQI. The questionnaire consisted of 10 single-choice questions, and each question was assigned a point value according to a Likert scale ranging from 0 to 3. The maximum score was 30 points. The higher the score, the more impacted the patient’s QoL. The reliability of the Polish version of the DLQI, expressed by a correlation coefficient, was 0.56. Its internal consistency was also very good, as confirmed by a high value of the Cronbach’s coefficient (0.90).

The BDI was applied in the full version of 21 items corresponding to all major depressive symptoms. The first 13 items pertain to the affective–cognitive symptoms of depression, including: pessimism, feeling of guilt, low mood, sense of failure, self-hate, loss of satisfaction, sense of punishment, loss of self-acceptance, crying for help, wish for death, irritability, social withdrawal, difficulties in decision-making. The other 8 items cover somatic symptoms, which may be present in depression: disturbed body image, disturbed sleep, fatigue, work inhibition, loss of appetite, loss of weight, somatic preoccupation, and lowered libido. There are four response options for each item and a score from 0 to 3 is given based on severity, with a total score of 0–39 grouping patients in groups of no depressive disorder (0–11), mild depression (12–26), moderate depression (27–37), severe depression (38–49) and very severe depression (50–60).

To analyze personality traits, we used the NEO-FFI scale. The questionnaire comprises 60 items used for self-assessment on a 5-point Likert scale. 

### 2.3. Statistical Analysis

The data collected in the spreadsheet were analyzed using Statistica (version 13 PL) and MedCalc (version 15.8 PL) software. Based on the results of the D’Agostino–Pearson test, the normality of the distribution of continuous variables was verified. Due to the non-normal distribution of continuous variables, non-parametric tests were used in the subsequent stages of statistical analysis. For the same reason, the median was used as a measure of data concentration, while data dispersion was presented using the interquartile range (IQR) and minimum–maximum range. The Mann–Whitney U test was used when comparing two independent groups regarding the distribution of a continuous variable. Statistically significant results of the comparisons mentioned above were presented using box-and-whisker plots. Dichotomized or categorical variables were expressed using absolute numbers and percentages. Comparisons of dichotomous or categorical variables between the groups were performed using the Fisher exact test (if two groups and categories of a variable were compared) or the Chi-square test with the Yates correction (if more than two categories of a variable were compared). In the case of comparisons of these types of variables, statistically significant results were presented using bar charts. The risk/chance of a specific clinical event occurring (depending on the presence of a given variant of the analyzed categorical variable) was estimated using the odds ratio test. In all tests used in the statistical analysis, results in which the alpha error (*p*) had values lower than 0.05 were interpreted as significant. In turn, results with a *p*-value in the range of 0.05–0.07 were considered to show a trend towards statistical significance.

## 3. Results

### 3.1. Characteristics and Comparison of the Study and Control Groups in Terms of Basic Demographic, Environmental and Educational Variables

The study included 30 patients (21 women and 9 men) with genodermatoses aged 16 to 79 (Table 1). Detailed characteristics of the study group are presented in Appendix A. The control group comprised 30 age- and sex-matched individuals without skin disorders. 

The study and control groups were balanced in terms of variables such as: gender dominated by women (70% vs. 70%; *p* = 0.7782), median age (36 years and 36.5 years, respectively; *p* = 0.8940), place of residence (urban residents constituted 60% and 83.3%, respectively; *p* = 0.0856) and education; in both groups, people with higher education predominated (percentages: 50% vs. 73.3%, respectively; *p* = 0.1987).

Our study group was diverse and consisted of many cases that differed genotypically and phenotypically, but with a common feature of persistent skin disease. In our study, 23 patients were classified as ichthyoses, 5 as keratodermas, 1 had ectrodactyly–ectodermal dysplasia clefting (EEC) syndrome [11], and 1 had erythrokeratoderma variabilis-like lesions. The confirmed mutations are presented in Appendix A.

### 3.2. Clinical Characteristics of the Study Group

The key clinical details of the study group are summarized in Table 1. Lesions on exposed body parts were noted in 53.3% of respondents, while scales on the entire body were observed in 46.7% of patients. The most frequently observed scales were of different colors and the median percentage of body surface area covered by lesions was 90% (min–max: 1–100%). The medians (and min–max ranges) of the scores obtained in the scales assessing skin itching, peeling and redness were 4.5 (1–10), 7 (1–10), and 5 (1–10), respectively. According to the VAS scale, the median (and min–max range) of pain values was 2 (0–8). Infiltration was noted in 43.3% of respondents. Increased sweating, no sweating and no sweating at all (hypohidrosis and anhidrosis) were observed in 33.3%, 33.3% and 10% of patients, respectively. Ectropion was present in 26.7% of respondents. Sleep disorders were recorded in 33.3% of patients. However, eyelid closure, ear plugging and hearing loss were noted in 56.7%, 43.4% and 20% of respondents, respectively. The overall performance status, measured using the Karnofsky scale, had a median of 90 points for the examined patients, with a range 25–100 points. That means that patients with median performances could carry on normal activities with minor disease symptoms. Disease-causing variants were found in all patients subjected to molecular testing. 

### 3.3. Quality of Life Assessed Using the Dermatology Life Quality Index 

A significantly higher incidence of reduced QoL according to the DLQI scale was observed in the study group compared to the control group (100% vs. 30%; *p* < 0.0001). Therefore, it was estimated that the risk of reduced QoL was significantly (131-fold) higher in the study group compared to the control group (OR = 138.1, 95%CI: 7.6–2501.3; *p* = 0.0009). Detailed comparisons revealed slightly, moderately, severely, or very severely reduced DLQI, observed significantly more frequently in the study group as compared to controls (33.3%, 23.3%, 36.7%, or 6.7% vs. 20%, 6.7%, 3.3% or 0%, respectively; Figure 1A). Moderately, severely, and very severely reduced QoL assessed according to the DLQI scale also occurred significantly more often in the study group compared to the control group (43.3% vs. 10%; *p* < 0.0001). Therefore, it was estimated that the risk of moderately reduced, severely, or very severely reduced QoL was significantly (18 times) higher in the study group compared to the control group (OR = 18, 95%CI: 4.4–74; *p* = 0.0001). Moreover, severely reduced and very severely reduced QoL according to the DLQI scale also occurred significantly more often in the study group compared to the control group (43.3% vs. 3.3%; *p* < 0.0008). It was estimated that the risk of severely or very severely reduced QoL was approximately 22 times higher in the study group compared to the control group (OR = 22.2, 95%CI: 2.7–184.8; *p* = 0.0042). Detailed data on the comparison of the QoL assessed using the DLQI in the study group and control are presented in Table 2.

### 3.4. Incidence and Level of Depression

In the study group, there was a higher prevalence of depression than in the control group (43.3% compared to 10%, *p* = 0.0086). Therefore, it was estimated that the risk of depression was approximately seven times higher in the study group compared to the control group (OR = 6.8, 95% CI: 1.7–27.7; *p* = 0.0067). Furthermore, severe depression was significantly more prevalent among study participants than controls (36.7% vs. 0%; *p* = 0.0016; Figure 1B). Detailed data regarding the comparison of the study and control groups in terms of the occurrence and level of depression assessed using BDI are presented in Table 3.

### 3.5. Comparison of the Study and Control Groups in Terms of the Personality Inventory

The study group and controls did not differ significantly in terms of personality inventory. Detailed data regarding the comparison of the study group and control in terms of personality inventory assessed using the NEO-FFI questionnaire are presented in Table 4.

### 3.6. The Relationship of Demographic, Social, Educational, and Clinical Variables with the Dermatology Life Quality Index in the Study Group

In patients with a significantly or very severely reduced QoL, a significantly higher median redness score was recorded compared to those in whom it was normal, or slightly or moderately reduced (medians, respectively: 6 vs. 4; *p* = 0.0019; Figure 2A). Moreover, a significantly higher incidence of severe or very severe reduction in QoL was noted in patients with sleep disorders compared to those who had no sleeping problems (rates: 80% vs. 25%, respectively; *p* = 0.0133; Figure 1C). None of the other assessed demographic, social, educational, and clinical factors were significantly associated with QoL. Detailed data on the association of demographic, social, educational, and clinical factors with QoL assessed by the DLQI are presented in Appendix A.

### 3.7. Relationship between Personality Inventory Assessed by NEO-FFI Questionnaire and Quality of Life Assessed by Dermatology Life Quality Index in the Study Group

In patients with a low or average level of agreeableness compared to those with a high level of agreeableness, a significantly more frequent occurrence of severely or very severely reduced QoL was observed (percentages: 100% or 75% vs. 28.6%, respectively; *p* = 0.0400; Figure 1D). None of the other parameters assessed using the NEO-FFI questionnaire were significantly correlated to the QoL. Detailed data on the relationship between personality inventory assessed using the NEO-FFI questionnaire and the QoL assessed using the DLQI in the study group are presented in Appendix A.

### 3.8. Relationship of Demographic, Social, Educational, and Clinical Variables with the Occurrence of Depression in the Study Group

Patients with depression had a significantly higher median skin itching score than those without depression (medians: 7 vs. 4, respectively; *p* = 0.0238; Figure 2B). Moreover, it was observed that patients with depression had a slightly higher median exfoliation score than those without depression (however, the result show a trend towards significance: medians: 9 vs. 5, respectively; *p* = 0.0518; Figure 2C). None of the other assessed demographic, social, educational, and clinical factors were significantly associated with depression (assessed according to the Beck questionnaire). Detailed data on the association of demographic, social, educational, and clinical factors with the occurrence of depression (Beck’s scale) are presented in Appendix A.

### 3.9. Relationship between Personality Inventory Assessed Using the NEO-FFI Questionnaire and Depression Evaluated with Use of Beck’s Scale in the Study Group

In patients with high levels of neuroticism compared to those with average or low levels of neuroticism, a significantly higher incidence of depression was observed (rates: 66.7% vs. 9.1% or 0%, respectively; *p* = 0.0067; Figure 1E). In turn, in patients with a low or average level of agreeableness compared to those with a high level of agreeableness, a significantly higher incidence of depression was noted (percentages: 100% or 75% vs. 28.6%, respectively; *p* = 0.0400; Figure 1F). None of the other parameters assessed using the NEO-FFI questionnaire were significantly associated with the occurrence of depression. Detailed data on the relationship between personality inventory assessed using the NEO-FFI questionnaire and depression in the study group are presented in Appendix A.

## 4. Discussion

Skin diseases significantly impact the daily activities and mental well-being of affected patients. Therefore, research into QoL makes a valuable contribution to dermatology. Our study demonstrates a reduction in QoL among all patients examined. QoL, as assessed by DLQI, was slightly reduced in 33.3% of suffering patients, moderately reduced in 23.3%, severely reduced in 36.7%, and very severely reduced in 6.7%. 

The study by Mazereeuw-Hautier et al. was the first work that investigated specific factors influencing the QoL of patients with ichthyosis, including mobility-limiting pain [12]. The study by Dreyfus demonstrates that pain, discomfort, and social aspects were most strongly correlated with the clinical severity of the disease. The highest DLQI scores were observed in patients reporting cutaneous pain, female patients and patients who lived alone [13]. Our study shows that patients with severely reduced QoL more frequently reported increased skin redness and sleep disorders. Enhanced skin redness may negatively influence social interactions, potentially resulting in increased psychological distress and a subsequent reduction in QoL. Additionally, the associated sleep disorders contribute to a cycle of fatigue and cognitive impairment, which intensifies irritability and further diminishes the QoL.

The available research shows that the QoL can also depend on the age and sex of the patients. According to Ganemo, the impact of the skin disease varies throughout a patient’s life, but is most pronounced in childhood [9,14]. However, in a study by Kamalpour et al., adults had significantly higher mean DLQI scores than children, and women had significantly higher mean DLQI scores than men (irrespective of age) [15]. In the study by Sun et al., male patients reported slightly lower median DLQI scores [10]. According to Gutiérrez-Cerrajero, one of the reasons for the higher DLQI scores for women is significant social pressure regarding their outward appearance [16].

Another significant aspect is the relationship between genodermatoses and the mental/emotional state of patients. In our study, depression was found in 43.3% of patients, while 36.7% of them had severe depression. The control group included 90% of individuals with no depression syndromes and 10% with mild depression. While the specific numbers may differ due to conditions such as the different demographics of the study group or different cultural circles, the trend visible in our study agrees with the literature. Cortés et al. described a group of 24 patients with lamellar ichthyosis aged 17–52 years. The BDI scores demonstrate that 20% of the respondents had no depression, 20% showed mild depression, while moderate and severe depression were observed in 30% and 30% of the patients, respectively. As for the healthy control group, 79% of the participants did not have depression, 4% had mild depression, 13% had moderate depression and 4% had severe depression [1]. Other studies have reached the same conclusions, showing that patients with different dermatological conditions have lower QoL and experience depressive and anxiety symptoms [3,17,18,19]. 

The most prevalent reason for such a trend may be the visibility of skin lesions, which make patients suffer from impaired self-image and relationships with other people. As for self-image, patients felt sad, lonely, discouraged, and angry. In terms of social contacts, patients mentioned inquisitive questions, staring, and lack of tact shown by others. Furthermore, patients were concerned about repulsion on the part of their potential or existing partner, which could lead to relationship withdrawal. All these factors can increase the risk of depression and anxiety. Therefore, all patients and their families should be offered life-long psychological care from healthcare providers (psychologist, dermatologist, nurses, etc.). Such assistance can include family therapy, self-management programs, group interviews and advice to parents regarding close physical contact with their affected infant. Moreover, it is important to inform patients and their families about various support groups operating in the country [20].

Our study found a significantly higher incidence of depression in patients with high levels of neuroticism compared to those with average or low levels of neurotoxicity. Neuroticism represents the tendency of an individual to experience psychological stress [21] and is the trait of the disposition to experience negative effects, including depression [22]. On the other hand, depression disorder, which pairs with lower self-esteem, can affect self-report responses on the neuroticism scale [21]. The research showed that neuroticism scores increased as clinical patients entered a depressive phase [23].

Our finding is also consistent with other evidence showing that neuroticism is associated with depression [24,25,26]. The study of Xia et al. showed that a large group of Chinese women with recurrent major depression disorder had significantly higher scores of neuroticism compared to the not-depressed control group [26]. Hirschfeld et al. found that patients that had not recovered from clinical depression after one year of depression diagnosis had higher scores of neuroticism when compared to participants whose symptoms had completely remitted [25].

Moreover, our study showed that patients with a low or average level of agreeableness had a significantly higher incidence of depression compared to those with a high level of agreeableness. Agreeableness is a dimension of interpersonal behavior; while individuals with high trait agreeableness are trusting and cooperative, low agreeableness is manifested in cynical, antagonistic behaviors [21]. Lower degrees of agreeableness can lead to greater perceived stress and depressive symptoms [27]. In their original study, Costa and McCrae did not find significant relationship between agreeableness and depression [21]. However, Merril et al. demonstrated that those people with high agreeableness had lower odds of developing depression, when compared to people with low agreeableness [28]. This study also indicates that, compared to individuals with low neuroticism, those with high neuroticism were more likely to develop depression. Thus, our findings on the link between neuroticism or agreeableness and depression in patients with ichthyosis are consistent with the wider literature on relationships between personality and psychological health.

Patients with genodermatoses, including disorders of keratinization, should receive the same level of care and consideration as those with psoriasis, acknowledging their need for specialized attention due to the potential for depression and reduced quality of life. Dermatologists should incorporate these findings into their practice by actively referring patients experiencing low mood to psychological services, addressing any accompanying conditions, and implementing preventive strategies to safeguard their well-being.

A primary limitation of our study is the sample size, which may restrict the generalizability of the findings. The limited sample size reflects the challenge of recruiting a large participant pool due to the extreme rarity of the conditions studied. Furthermore, the distribution of patients with varying dermatological conditions was not uniform; individuals with ichthyosis vulgaris and lamellar ichthyosis represented 63% of the study population. The study acknowledges the potential bias in evaluating a heterogeneous group of genodermatoses, which vary in clinical severity. Despite this variability, it is essential to note that a decline in QoL was a universal finding among all participants. Therefore, it would be inappropriate to rank the conditions by severity based on subjective experiences, as each individual’s perception of their disease’s impact is valid and significant. 

Conversely, the strengths of this study lie in its novel exploration of the correlation between depression, QoL, and psychological traits. The diversity of the study cohort, encompassing eight different types of genodermatoses across a broad spectrum of ages, adds a valuable dimension to the existing research.

## 5. Conclusions

Genodermatoses including ichthyosis significantly diminishes the QoL for patients, with nearly half of our participants experiencing a severe reduction in their overall well-being. This is often attributed to active skin lesions and sleep difficulties. 

Ichthyosis is correlated with a 131-fold higher risk of depression than for people without skin disorders. Moreover, patients with a low or average level of agreeableness and those with high levels of neuroticism had a significantly higher incidence of depression. Lastly, concerning personality traits, individuals with ichthyosis do not exhibit significant differences from healthy individuals, irrespective of the screening tool employed.

## Figures and Tables

**Figure 1 jcm-13-01624-f001:**
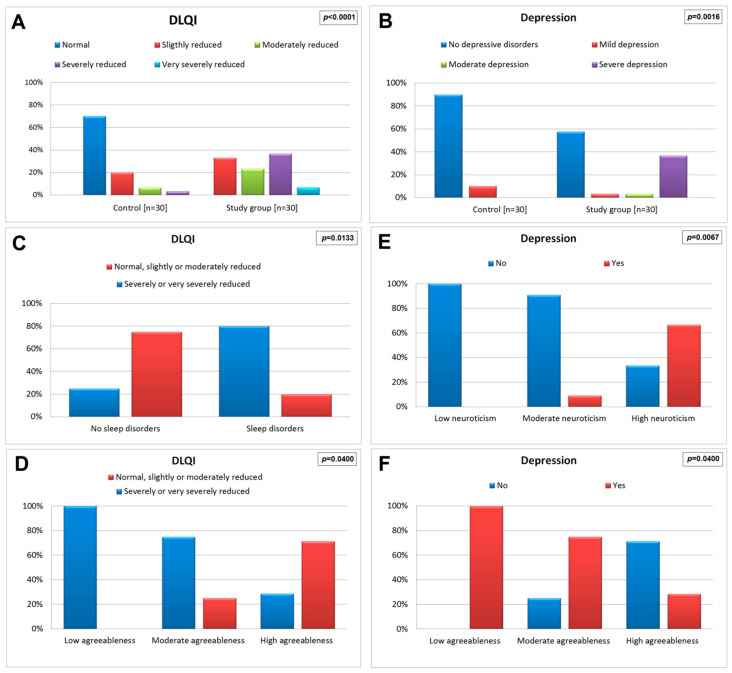
Bar graphs demonstrating the comparison of: quality of life (**A**) or the occurrence and severity of depression (**B**) in the study group and the controls; quality of life depending on sleep disorders (**C**) or agreeableness level (**D**) in the study group; occurrence of depression depending on neuroticism (**E**) or agreeableness (**F**) level in the study group.

**Figure 2 jcm-13-01624-f002:**
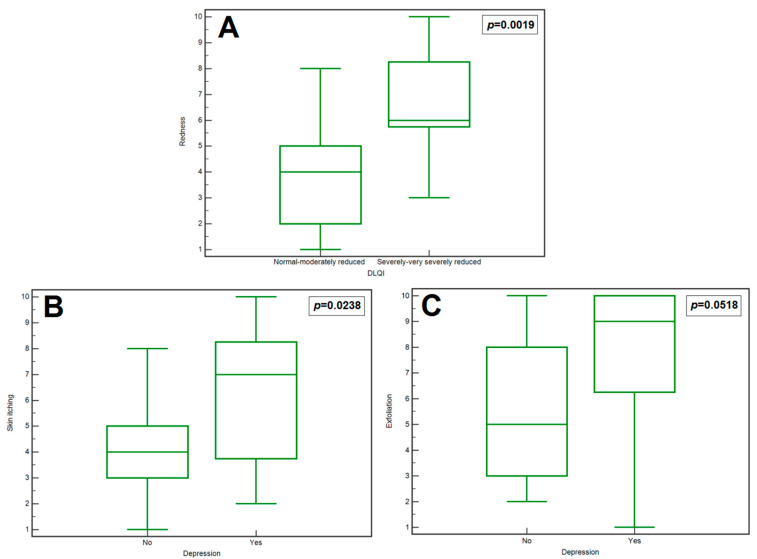
Box–whisker graph demonstrating the comparison of: the redness depending on DLQI level in the study group (**A**); the skin itching (**B**) or the exfoliation (**C**) depending on depression occurrence in the study group.

**Table 1 jcm-13-01624-t001:** Patient characteristics.

Patient Characteristics	Values	Study Group (n = 30)
Sex	Men	9 (30%)
	Women	21 (70%)
Age (years)	median [IQR] (Min–Max)	36 [26–47] (16–79)
Genodermatosis type	Autosomal Recessive Congenital Ichthyosis (ARCI)	1 (3.3%)
	Ectrodactyly–ectodermal dysplasia clefting syndrome (EEC)	1 (3.3%)
	Erythrokeratodermia variabilis (EV)-like changes	1 (3.3%)
	Erythrodermic ichthyosis	2 (6.7%)
	Ichthyosis with confetti	1 (3.3%)
	Lamellar ichthyosis	9 (30%)
	Keratoderma palmoplantare	5 (16.7%)
	Ichthyosis vulgaris	10 (33.3%)
Scales on exposed parts of the body	No	14 (46.7%)
	Yes	16 (53.3%)
% of the body surface covered in lesions	Median [IQR] (Min–Max)	90% [10–100%] (1–100%)
The whole body is covered in scales	No	16 (53.3%)
	Yes	14 (46.7%)
Color of scales	Transparent	1 (3.3%)
	White	4 (13.3%)
	Yellow	4 (13.3%)
	Brown	5 (16.7%)
	Beige	6 (20%)
	Multicolored	10 (33.3%)
Skin itching	Median [IQR] (Min–Max)	4.5 [3–7] (1–10)
Redness	Median [IQR] (Min–Max)	5 [3–6] (1–10)
Exfoliation	Median [IQR] (Min–Max)	7 [4–9] (1–10)
Infiltration	No	17 (56.7%)
	Yes	13 (43.3%)
Pain (VAS)	Median [IQR] (Min–Max)	2 [1–5] (0–8)
Level of sweating	Absolutely no sweating	3 (10%)
	No sweating	10 (33.3%)
	Normal sweating	7 (23.3%)
	Increased sweating	10 (33.3%)
Ectropion	No	22 (73.3%)
	Yes	8 (26.7%)
Sleep disorders	No	20 (66.7%)
	Yes	10 (33.3%)
Inability to fully close the eyelids	No	17 (56.7%)
	Yes	13 (43.4%)
The feeling of ear plugging	No	17 (56.7%)
	Yes	13 (43.4%)
Hearing loss	No	24 (80%)
	Yes	6 (20%)
Performance status (on the Karnofsky scale) scale of ability	Median [IQR] (Min–Max)	90 [60–100] (25–100)

**Table 2 jcm-13-01624-t002:** Comparison of study group and control in terms of quality of life assessed using the Dermatology Life Quality Index.

Variable: DLQI	Study Group (n = 30)	Controls (n = 30)	*p*
Normal	0 (0%)	21 (70%)	<0.0001 *
Slightly reduced	10 (33.3%)	6 (20%)	
Moderately reduced	7 (23.3%)	2 (6.7%)	
Severely reduced	11 (36.7%)	1 (3.3%)	
Very severely reduced	2 (6.7%)	0 (0%)	
Normal	0 (0%)	21 (70%)	<0.0001 *
Reduced	30 (100%)	9 (30%)	
Normal, slightly reduced	10 (43.3%)	27 (90%)	<0.0001 *
Moderately reduced,	20 (56.7%)	3 (10%)	
Severely reduced, very severely reduced			
Normal, slightly reduced, moderately reduced	17 (56.7%)	29 (96.7%)	0.0008 *
Severely reduced, very severely reduced	13 (43.3%)	1 (3.3%)	

* Statistically significant result.

**Table 3 jcm-13-01624-t003:** Comparison of the study group and control in terms of the occurrence and level of depression assessed using Beck’s scale.

Variable: Depression	Study Group (n = 30)	Control (n = 30)	*p*
No depressive disorders	17 (56.7%)	27 (90%)	0.0086 *
Depression	13 (43.3%)	3 (10%)	
No depressive disorders	17 (56.7%)	27 (90%)	0.0016 *
Mild depression	1 (3.3%)	3 (10%)	
Moderate depression	1 (3.3%)	0 (0%)	
Severe depression	11 (36.7%)	0 (0%)	

* Statistically significant result.

**Table 4 jcm-13-01624-t004:** Comparison of the study group and control in terms of personality inventory assessed using the NEO-FFI questionnaire.

Personality Inventory	Level	Study Group (n = 30)	Control (n = 30)	*p*
Openness to experience	Low	2 (6.7%)	0 (0%)	0.3292
	Moderate	8 (26.7%)	10 (33.3%)	
	High	20 (66.7%)	20 (66.7%)	
Neuroticism	Low	1 (3.3%)	2 (6.7%)	0.4158
	Moderate	11 (36.7%)	15 (50%)	
	High	18 (60%)	13 (43.3%)	
Agreeableness	Low	1 (3.3%)	2 (6.7%)	0.8187
	Moderate	8 (26.7%)	7 (23.3%)	
	High	21 (70%)	21 (70%)	
Extroversion	Moderate	9 (30%)	7 (23.3%)	0.7703
	High	21 (70%)	23 (76.7%)	
Scrupulousness	Low	0 (0%)	1 (3.3%)	0.1949
	Moderate	9 (30%)	4 (13.3%)	
	High	21 (70%)	25 (83.3%)	

## Data Availability

The data that support the findings of this study are available from the corresponding author, upon reasonable request.

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
