# Peer review of "The Depressiveness, Quality of Life and NEO-FFI Scale in Patients with Selected Genodermatoses"

_jcm, 2024, doi:10.3390/jcm13061624_

Round 1

Reviewer 1 Report

Comments and Suggestions for Authors

This research studies the relationship  between depressive symptoms, quality of life (QoL), and personality traits in individuals diagnosed with  genodermatoses. The study cohort comprised 30 patients with genoderma and a healthy control group. Standardized survey question DLQI, BDI, and NEO-FFI were employed for assessments.  The findings indicate a significantly elevated risk of  severely reduced QoL in patients with genodermatosis. This is a novel and interesting study in this field.

Author Response

Authors:  Thank you for your time and the constructive feedback provided in your review. We appreciate the effort and thoughtfulness you put into evaluating our work. 

Reviewer 2 Report

Comments and Suggestions for Authors

This study demonstrates a substantial correlation between genodermatoses and the mental health of affected individuals, and underscores the consideration of psychological factors in the management of hereditary skin disorders. The study is well-designed with scientific writing. Some suggestions are listed as follows.

1. Please give the full name of DLQI, BDI, and NEO-FFI in the abstract.

2. Please add a paragraph to describe the impact of this study on clinical practice. Are there any suggestions for the dermatologists to approach these patients?

Author Response

This study demonstrates a substantial correlation between genodermatoses and the mental health of affected individuals, and underscores the consideration of psychological factors in the management of hereditary skin disorders. The study is well-designed with scientific writing. Some suggestions are listed as follows.   Authors: We appreciate the time you've taken and the valuable input you've provided in your review.  

1. Please give the full name of DLQI, BDI, and NEO-FFI in the abstract.

Authors: We have included the full names as requested.

2. Please add a paragraph to describe the impact of this study on clinical practice. Are there any suggestions for the dermatologists to approach these patients?   Authors: We have added a paragraph to the discussion section addressing these points.

Reviewer 3 Report

Comments and Suggestions for Authors

This study underscores a substantial correlation between genodermatoses and the mental health of affected individuals, emphasizing the indispensable consideration of psychological factors in the management of hereditary skin disorders. However, it is imperative to acknowledge the primary limitation of our study, namely the small sample size, attributed to challenges in recruiting participants due to the rare nature of the studied conditions.

Furthermore, the study acknowledges the challenges posed by the heterogeneous nature of genodermatoses and the limitations in evaluating their clinical severity.

While acknowledging the authors' efforts in conducting this study, it's essential to recognize that the information presented might not be entirely novel. The study contributes to the existing body of literature by applying standardized survey questionnaires in a novel context, shedding light on the intricate relationship between depressive symptoms, quality of life (QoL), and personality traits in individuals with genodermatoses.

Although not presenting entirely new information, the study offers valuable insights and methodological advancements to dermatological and psychological health research.

Author Response

Authors: We are grateful for the time you have dedicated and the insightful feedback you provided in your review. As mentioned in our discussion, the small sample size is the primary limitation of our work, which is influenced by the rarity of the disease. Please let us know if you have any other suggestions or comments about our paper.

Round 2

Reviewer 2 Report

Comments and Suggestions for Authors

The authors have addressed the comments. 

Reviewer 3 Report

Comments and Suggestions for Authors

I have no additional comments to add. In my opinion, the article is suitable for publication.